# Design and Screening of New Lead Compounds for Autism Based on QSAR Model and Molecular Docking Studies

**DOI:** 10.3390/molecules27217285

**Published:** 2022-10-26

**Authors:** Yongjian Chen, Kang Ma, Peilong Xu, Hongzong Si, Yunbo Duan, Honglin Zhai

**Affiliations:** 1School of Public Health, Qingdao University, Qingdao 266071, China; 2School of Basic Medicine, Ningxia Medical University, Yinchuan 750004, China; 3New Fibrous Materials and Modern Textile State Key Laboratory, Qingdao University, Qingdao 266071, China; 4Quantum Institute, Queenland Montessori College, Surrey, BC V3Z 0T3, Canada; 5Department of Chemistry, Lanzhou University, Lanzhou 730000, China

**Keywords:** antipsychotics, autism, azinesulfonamides, cyclic amine derivatives

## Abstract

The purpose of the present study aims to develop a satisfactory model for predicting pro-social and pro-cognitive effects on azinesulfonamides of cyclic amine derivatives as potential antipsychotics. The three dimensional-quantitative structure affinity relationship (3D-QSAR) study was performed on a series of azinesulfonamides of cyclic amine derivative using comparative molecular similarity indices analysis (CoMSIA). The best statistical model of CoMSIA q2, r2, SEE and F values are 0.664, 0.973, 0.087, and 82.344, respectively. Based on the model contour maps and the highest activity structure of the 43rd compound, serial new structures were designed and the 43k1 compound was selected as the best structure. The dock results showed a good binding of 43k1 with the protein (PDB ID: 6A93). The QSAR model analysis of the contour maps can help us to provide guidelines for finding novel potential antipsychotics.

## 1. Introduction

Autism is a kind of syndrome caused by abnormal neuropsychological function, which leads to serious problems in communication, social communication, and behavior. Researchers have defined autism as a neuro-developmental disorder that autism affects the ability to interact and communicate with other people [1].

At present, autism is a common psychological disease in children [1]. According to a 2016 report by the National Center for Health Statistics, the prevalence of autism among children in the United States is 2.2%, while the prevalence of autism in China is about 1 percent. Among them, there are about 20% of new cases per year and most of them are children [2]. Autism can cause a life quality decrease, gastrointestinal symptoms, sleep problems, and social problems [3]. And it will affect the child’s development and life in the future. Drugs are the main way to treat this disease. Nowadays, it has been found that most antipsychotic drugs can bind to dopamine D2 receptors (D2Rs) in the mesolimbic and nigrostriatal brain areas [4]. Although these drugs can alleviate symptoms, they also bring forth side effects, such as induction of extrapyramidal symptoms (EPS) and hyperprolactinemia. Therefore, second-generation antipsychotics with a high affinity for various serotonin receptors were found. Since the 5-HT2A/D2 has higher receptor affinity, 5-HT1AR activation contributes to the antidepressant and anti-anxiety properties of antipsychotics. The blockade of 5-HT6Rs has a novel mechanism to treat cognitive impairment in schizophrenia [5]. In this regard, antagonists and inverse agonists of 5-HT6R display precognitive actions in preclinical and clinical settings [6]. Preclinical data demonstrate that the 5-HT7R antagonist SB-269970 exerts pro-social effects [7]. To find novel and effective structures, artificial intelligence can be used for new drug findings. 

The Quantitative Structure-Affinity Relationship (QSAR) method has been used for new drug finding and designing. There are a few studies of QSAR has been applied to find antipsychotics. Benzamide antipsychotics were classified as second-generation antipsychotics, the QSAR model showed the structures of 5-sulfonyl on the benzene ring and N-ethyl-2-aminomethyl pyrrolidine played a role in antipsychotic activity [8,9]. The biological activities of psychiatric drugs were evaluated by QSAR models [10]. This study found that 2-trifluoromethyl-phenothiazine dihydrochloride and 2-trifluoromethyl-phenothiazine hydrochloride derivatives exhibited an influence on the frequency of necrotic effect on lymphocytes in comparison with fluphenazine [11]. Based on the works, the QSAR model can aid to understand the mechanism of compound interaction with targets, the main important structures that can influence the activities. At present, there are no other QSAR studies for azinesulfonamides of cyclic amine derivatives.

Higher activity and fewer side effects drugs are needed in clinics, however, traditional drugs finding are time and cost-consuming. At present, artificial intelligence is used in nearly most drug development, such as in structure design, bioinformatics, property prediction, and so on. Therefore, computational methods have been extensively used for drug discovery and development. 

QSAR method is a mathematical relationship linking chemical structure and pharmacological affinity or other properties. QSAR has been applied widely to drug discovery which can predict affinity-related characteristics of drug candidates [12,13,14]. The QSAR techniques will be used in a vast area of functional chemicals such as pharmaceuticals, agrochemicals, flavor, perfumeries, analytical reagents, solvents, and household chemicals. It also can be applied to the chemicals of health benefits, the chemicals responsible for environmental hazards, and the chemicals in the industrial/laboratory processes [15,16]. In this study, CoMFA and comparative molecular similarity indices analysis (CoMSIA) were used to build the QSAR model. However, CoMFA only has stereoscopic and electrostatic force fields. CoMSIA provides five force fields, including stereoscopic and electrostatic force fields, so we only used the CoMSIA technique in our work. CoMSIA as one of the QSAR methods was applied to develop a satisfactory model based on azinesulfonamides of cyclic amine derivatives as potential antipsychotics with pro-social and pro-cognitive effects. Meanwhile, the developed model with visualize structural information was used to design more structures. CoMSIA can provide stereoscopic, electrostatic, hydrophobic, hydrogen bond donors and hydrogen bond recipients force fields. The 43rd compound has the best Ki value, so it was chosen as the template molecule. We can modify 43rd structure at the five force fields position. Lots of new structures will be found. Then all the new compounds will have been predicted pKi values by our QSAR model. The lowest Ki value structure is the best one which was selected to do binding with protein. According to the QSAR model, the new structure 43k1 contained the hydrophobic group benzene. And the acceptor group F was used to modify the structure, two H-bonds were found between F and residues ASP-155 and SER-159. Therefore, the 43k1 compound has a good binding with protein (PDB code: 6A93).

## 2. Results and Discussion

### 2.1. Alignment Commune Statistics 

It is important to select an appropriate structural alignment of the compounds in 3D-QSAR analysis. There are two different alignment techniques, one is ligand-based alignment and the other is receptor-based alignment. In this study, the former was adopted (Figure 1). In this approach, all compounds were aligned to the most potent compound 43 in the Sybyl package because it demonstrated potential anti-autism activity with pKi of 5 nM and provided a new scaffold for drug development and the alignment moiety here was the common structure of each molecule (Figure 1).

Stereoscopic field, static electricity, hydrophobic, hydrogen bond donors, and hydrogen bond recipients were applied to analyze the CoMSIA model. These five molecular fields are the best statistical results of the CoMSIA model. Based on five fields and 15 different models were generated. Among them, 5 field parameters were used in model 15. It was the best model with superiority in all aspects of analysis. The 15th model is the best one. Of the 15th model, the q2, r2, SEE, and F values were 0.664, 0.973, 0.087, and 82.344, respectively. The hydrogen bond donor has the maximum contribution in this model. At the same time, CoMFA analysis also was finished, r2, q2, F values, and SEE were 0.973, 0.629, 82.352, and 0.087, respectively.

### 2.2. Validation of CoMSIA Model

Bootstrapping analysis for 100 runs and Y-randomization performed 10 times were applied to measure the bias of the original calculations and evaluate the robustness and the statistical significance of the derived models. In the former, an average r2 boot value of 0.986 (higher than r2) and a SEE boot value of 0.062 (lower than SEE) were calculated. In the latter, the q2 (0.558 to 0.721) and the $R_{p}^{2}$ value of 0.537 demonstrated that the derived model was not affected by chance correlations. The external predictive ability of the CoMSIA model was assessed by predicting the affinity of the test set. The statistic parameters suggested that the derived model had a good predictive ability. The experimental predicted activities and the residual values were listed in Table 1. Figure 2 showed a good relationship between predicted pKi values and experimental values. From Table 1, we can find that the 60th compound has the groups of 5-Isoquinolyl benzo[d]isoxazole and enant R, the 72nd and 53rd compounds have the structures of 4-Isoquinolyl benzo[d]isothiazole and enant S. To our knowledge, chemical structure determines chemical properties. QSAR model shows the relationship of structure with properties. Therefore, for the different structural features of the 60th, 72nd, and 53rd compounds compared with the others, the predicted pKi values have a larger variation (Figure 2).

### 2.3. CoMSIA Map

The CoMSIA model provides us with important information, which can show the relationship between the structures and biological affinity of compounds. We took the most affinity molecule 43rd as the reference molecule shown in Figure 3, and the three-dimensional color outline was constructed according to the best CoMSIA model. With the outline diagram, if the corresponding groups of the compounds were changed, the affinity of the compounds will change with the structure changing. Meanwhile, the contour maps, constructed the model by the field type StDev⁄Coeff (the standard deviation and the coefficient) with default values of 80% favored and 20% disfavored contributions. The contour maps help to identify the important regions where variation in the steric, electrostatic, hydrophobic, hydrogen bond donor, and acceptor fields around the compound [17]. 

The three-dimensional profile is shown in Figure 3A. The group of 2,3-dihydrobenzo[b][1,4]dioxine of 43 compounds falls into a sterically favorable green contour, suggesting that the steric bulk in this position is crucial for structure affinity relationships. The co-localization of the 2,3-dihydrobenzo[b][1,4]dioxine group within the green contour maps suggests the importance of steric bulk at this position. The two green contour maps placed onto the meta and para positions of the ring indicate that steric bulk at these positions is important for affinity. These two green contour maps were found to be colocalized with the blue contour map indicating a favorable effect of the steric bulk with high electron density containing groups. The small green contour maps around the ring suggest the sterically favorable region. Yellow contour map indicates that occupancy of these sterically unfavorable maps which have a negative effect on affinity. Figure 3B shows the electrostatic field diagram, in which the blue and red areas represent the positive and negative favorable areas, respectively. Figure 3B reveals the electrostatic contour maps for the CoMSIA analysis of affinity. The large blue contour maps indicate that the electropositive potential in these rings increases affinity. The electrostatic potential is more positive and thereby leads to compounds with enhanced affinity potencies. The red contour maps around the positions of the ring indicate that electronegative groups at these positions would decrease affinity. Compound 43 with 2,3-dihydrobenzo[b][1,4]dioxine substituent is more corresponding (electropositive) substituent.

Several small red contour maps around the ring suggest the importance of electronegative groups. The CoMSIA electrostatic contour map for affinity is depicted in Figure 3B. The small and large blue contour maps were located on the para positions. This finding indicated that the simultaneous presence of electron-withdrawing groups at the para positions of the 2,3-dihydrobenzo[b][1,4]dioxine was essential for potent affinity. Figure 3C showed the hydrophobic profile. Adding hydrophobic groups near the yellow profile is beneficial to increase the affinity of the compound, while the white profile indicated that reducing hydrophobic groups could increase the affinity of the compound. Otherwise, it will reduce the affinity of the compound. The profiles of hydrogen donors and hydrogen acceptors were shown in Figure 3D,E. The cyan and purple outlines provide useful sites for hydrogen donors. Magenta and red outlines indicated the favorable and unfavorable regions of hydrogen receptors for the affinity of compounds, respectively. Based on the map, new structures were designed, and a good activity for antipsychotic inhibitors has been screed. 

### 2.4. Topomer CoMFA Analysis

Before submitting the job for Topomer CoMFA analysis, each of the training set structures was broken into two sets of fragments. The two Topomer CoMFA model was obtained by changing the segmentation methods by using a training set. The q2 and r2 of the were 0.658 and 0.916. For a reliable predictive model, the q2 should be >0.5, the models are also obtained from both CoMFA and CoMISA, the q2 and r2 of the CoMFA model were 0.629 and 0.973, the q2 and r2 of CoMISA model were 0.664 and 0.973. Because Topomer CoMFA and CoMFA analysis results of q2 were lower than the CoMISA model. Especially topomer CoMFA and CoMFA only provide stereoscopic and electrostatic force fields, which were insufficient to analyze the design of new structures. Therefore, in this work, we only used the CoMISA model for new compound findings.

### 2.5. Design of New Compounds and Prediction of Their Activity

3D-QSAR has been widely used in the design and prediction of new compounds. Based on the above analysis, we obtained the main factors affecting affinity and added the corresponding groups based on the tips in each section of the contour map. From Table 2, the 15th model statistical parameters of q2, r2, F value, and SEE were 0.664, 0.973, 82.344, and 0.087 respectively. For the higher q2 and r2 values and lower SEE values, this model is the best one in our study. Since the structure of 43rd has the lowest Ki value, 43rd structure was modified according to the five force fields position of the 15th model. Several serials of new structures can be constructed. All new compounds were predicted pKi values by our QSAR model. Free combination, compound modification, and optimization selected 12 molecules higher than compound 43 (43a–43k). The pKi predictions for these newly designed compounds using CoMSIA are shown in Table 3. The lowest Ki value structure of 43k1 is the best one which was selected to do binding with protein. The structure 43k1 contained the hydrophobic group benzene. H-bond acceptor group F also appeared in the 43k1 structure. The docking result showed that two H-bonds were found between F and residues ASP-155 and SER-159. Therefore, the 43k1 compound has a good binding with protein (PDB code: 6A93).

### 2.6. Molecular Docking Results

Molecular docking is a good demonstration of the interaction between small molecules and proteins. The affinity and total score of our modified 43k1 compound is an ideal level, so we analyzed the 43k1 compound. By docking, we found that the new compound formed a hydrogen bond with the ASP-155 residue of the protein, as well as a hydrogen bond with the SER-159 residue. At the same time, the hydrophobic formed surrounds 43k1, but the electrostatic or steric effects are not found. The result shows that the 43k1 compound has a good binding with the polymer proteins (Figure 4A,B). To validate the 43k1 compound, the 43rd compound was also docked with protein (PDB code: 6A93). The docking score was 8.203 (Figure 4C), while the 43k1 compound dock score was 11.337 (Table 3). The interaction of the compound with the protein is important. There is no H-bond between the 43rd compound and the protein. Therefore, the 43k1 compound dock result and affinity prediction are very satisfactory.

## 3. Materials and Methods

### 3.1. Dataset for Analyses

In our study, we cited literature that includes 51 structures. Among them, 6 drugs did not share a common skeleton with the other 45 structures, so we only selected 45 common skeleton compounds to develop the model [18]. The Ki values in the nanomole (nM) range were converted to the molar value range and then to its logarithmic scale (pKi = −logKi + 9) used for subsequent QSAR analysis to reduce the skewness of the data set. The structures and biological affinity values of 45 compounds were shown in Table 1. The dataset was randomly divided into a training set of 34 compounds and a test set of 11 compounds in the approximate ratio of 4:1. Training set was used for 3D-QSAR model construction and the test set was used for model validation of the established models as extra independent samples in the QSAR analysis.

### 3.2. Optimization of Structures 

We built all the selected compounds using ChemDraw. We then put the constructed compounds into Sybyl software (SYBYL-X-2.1.1) for optimization and molecular modeling. Tripos force field and Powell gradient algorithm with a convergence criterion of 0.05 kcal/mol were used to carry out structural energy minimization and the minimized structures were used for CoMSIA as the initial conformation. 

### 3.3. Conformational Sampling and Alignment 

In 3D-QSAR, compound alignment is based on both mating and subject alignment. In this study, we chose a mating-based alignment. We selected the 43rd compound as a template molecule to align all compounds to because the 43rd compound has a high affinity with 5-HT1A in all compounds.

### 3.4. CoMSIA Model Study

CoMSIA in Sybyl was used for the model study. CoMSIA is one of the most reliable tools for studying 3D-QSAR because of its feature in avoiding singularities at the atomic positions arising from the Lennard-Jones and dramatic changes of coulomb potential fields. The CoMSIA descriptors were calculated by a 3D cubic lattice with a grid spacing of 2 and extending 4 units beyond the aligned molecules in all directions. The partial least squares (PLS) analysis was used to correlate the CoMSIA fields to the pKi values in order to generate a statistically significant 3D-QSAR model performed in two stages. Firstly, a leave-one-out (LOO) cross-validation analysis was carried out to determine the optimum number of components (ONC) and cross-validated correlation coefficient (q2). Then, non-cross-validation analysis was performed using the ONC to generate the final PLS regression models for CoMSIA. The non-cross-validation results were evaluated by several statistical parameters including the non-cross-validated correlation coefficient (r2), the standard error of estimate (SEE), and the F value. To further evaluate the robustness and the statistical significance of the derived, the validation model must be finished.

The CoMSIA model was assessed by predicting the affinity of 11 compounds of the test set and the external predictive correlation coefficient $R_{ext}^{2}$ was computed using the following equation:


(1)
$$R_{ext}^{2}=1-PRESS/SD$$


The SD is the sum of the squared deviations between the biological affinity of compounds in the test set and the mean biological affinity of training molecules, and the predictive of squares (PRESS) denotes the sum of square deviation between the predictive and experimental activities of the test molecules. In addition, $R_{m(overall)}^{2}$ was introduced and computed by the following equation to further validate the predictive ability of the model:


(2)
$$R_{m(overall)}^{2}=R^{2}*\left(1-\sqrt{R^{2}-R_{0}^{2}}\right)$$


In the equation, $R^{2}$ and $R_{0}^{2}$ are squared correlation coefficients between the observed and predicted activities of the test set with and without intercept, respectively. The parameter determines the extent of the range of predicted affinity values for the whole dataset and the observed affinity. Those values above including $R_{ext}^{2}$ and $R_{m(overall)}^{2}$ should be higher than 0.5 for a satisfactory model [19,20].

### 3.5. Topomer CoMFA

Topomer CoMFA is a rapid fragment-based three-dimensional quantitative structure-activity relationship (3D-QSAR) method. Unlike traditional CoMFA, the Topomer CoMFA does not require subjective alignment of 3D ligand conformers and uses automatic alignment rules, so analysis is faster. The steps of the Topomer CoMFA are as follows:(1)Split the 3D molecular structure into fragments containing common features, open valence bonds, or linkages.(2)Align each segment based on overlapping parts to provide an absolute orientation of any segment.(3)Calculate the steric and electrostatic fields of the top-aligned segments.(4)Use PLS regression to build the model and the jackknife test to evaluate the model.

The r2 and q2 were used to evaluate the Topomer CoMFA models. The cutoff values of r2 and q2 are 0.8 and 0.5, respectively. The optimum model was determined by the highest q2, and the validity of the model depends on the r2 value.

### 3.6. Molecular Docking 

Through molecular docking, we could study the molecular interaction mechanism and the appropriate binding mode. Molecular docking was carried out in the software Sybyl. We searched the RSCB protein database (PDB code: 6A93) and found that the structure of the ligand in 6A93 is similar to our selected 43rd compound. First, we modified the protein which was obtained from the PDB database. We introduced the protein into Sybyl software, removed the redundant water molecules, and extracted the ligands. Polar hydrogens and united atom Kollman charges were assigned for the receptor. Based on the automatic ligand model, the proposed binding affinity pocket can be created in the Sybyl package, where the ligand can be installed and potentially interact. The ProtoMol bloat and ProtoMol threshold parameters were set as default values of 0 and 0.5, respectively. Based on the PDB 6A93, this protein has the ligand, we created the activity pocket by the ligand structure interacting with the amino acid residue of the protein. We kept the top 10 conformations of each ligand and ranked them according to the total docking score. The best conformation of the ligand was analyzed by binding interaction. 

## 4. Conclusions

In our study, molecular docking has been employed to identify a potential binding mode for these inhibitors at the crystal structure of the 5-HT2AR affinity site and the best-docked conformation of molecule 43 was used as a template for alignment. The CoMSIA model showed a higher q2 value of 0.646 and an r2 value of 0.973, this model has a good predictive ability. The CoMSIA model provides five fields of steric, electrostatic, hydrophobic, hydrogen donors, and hydrogen acceptors modification that can improve the affinity ability. Finally, a new structure of 43k1 was added to the hydrophobic group benzene based on Figure 3C. And Figure 3E, the acceptor group F was used to modify the structure, two H-bonds were found between F and residues ASP-155 and SER-159. The docking result and affinity prediction are very satisfactory. Therefore, the hydrophobic group and H-bond acceptor group are modified and led to azinesulfonamides of cyclic amine derivatives which can improve the affinity. Then the new structures also have good binding with protein. The better-predicted model plays a key role in drug design and screening.

## Figures and Tables

**Figure 1 molecules-27-07285-f001:**
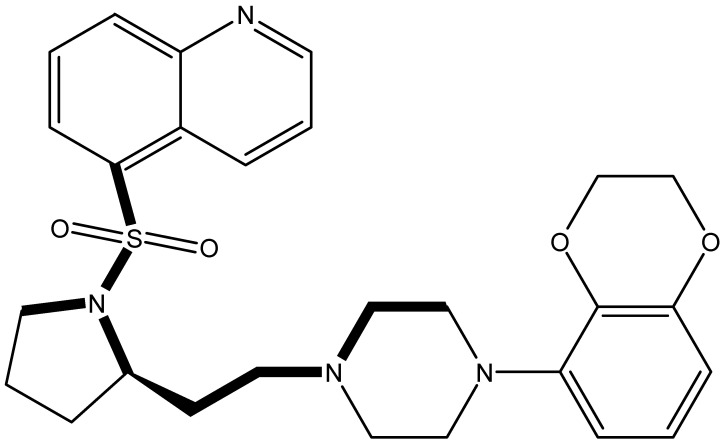
The alignment of all compounds in the dataset, compound 43 was used as the template and the common substructure (shown in bold) for the alignment of all compounds.

**Figure 2 molecules-27-07285-f002:**
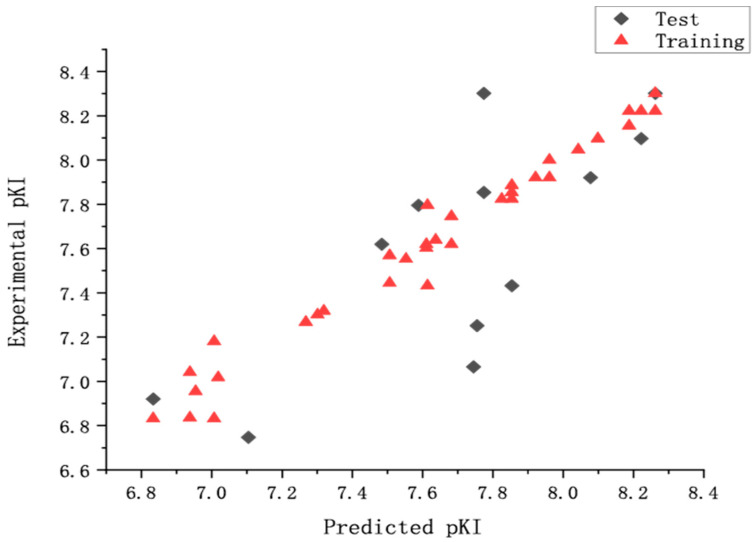
The predicted and the experimental pKi values (nM) of CoMSIA model.

**Figure 3 molecules-27-07285-f003:**
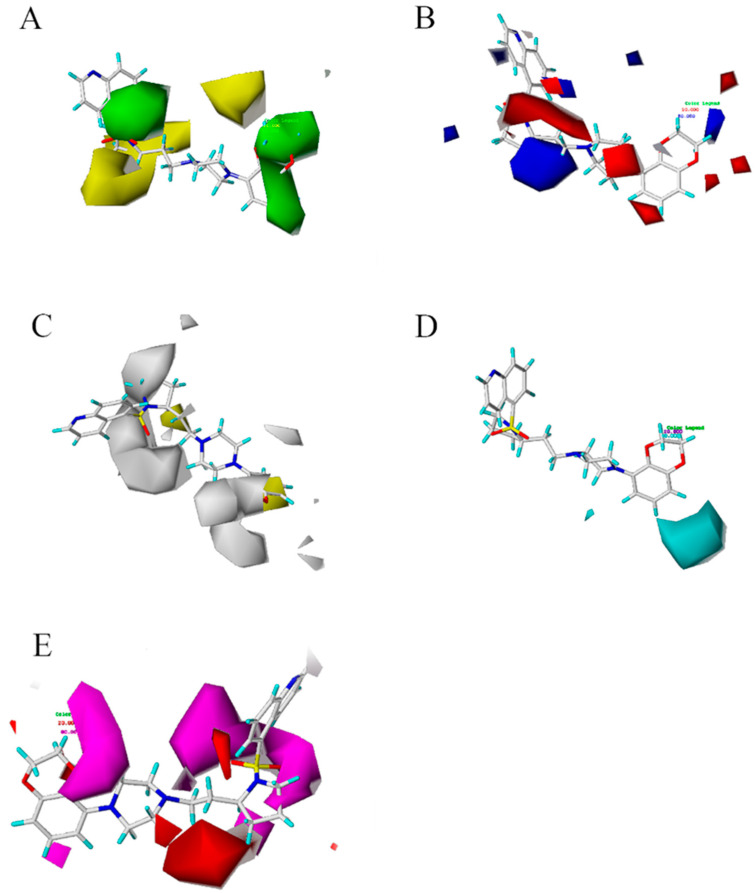
Contour maps of CoMSIA model based on 43rd structure. (**A**) is the steric field diagram, favorable (green) and unfavorable (yellow). (**B**) is the electrostatic field diagram, positive (blue) and negative (red) areas. (**C**) is the hydrophobic profile diagram, favorable (yellow) and unfavorable (white). (**D**) is the hydrogen donors diagram, favorable (cyan). (**E**) is the hydrogen acceptor diagram, favorable (magenta) and adverse (red) hydrogen receptor areas.

**Figure 4 molecules-27-07285-f004:**
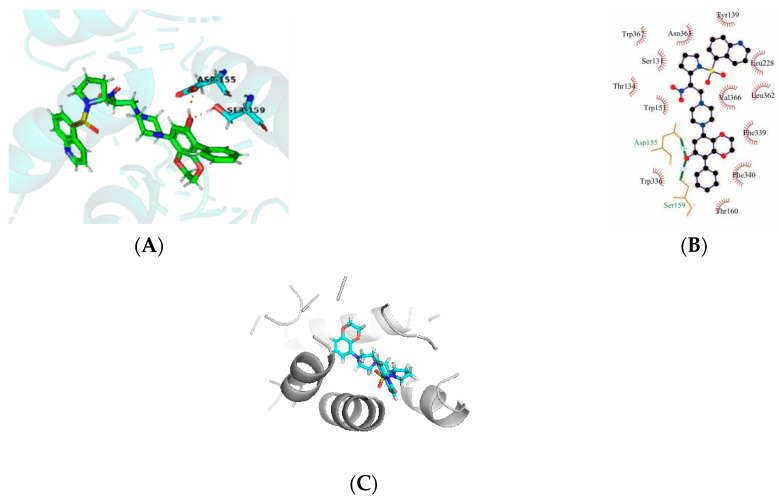
The molecular docking of newly designed compound 43k1 and compound 43 with protein 6A93. (**A**,**B**) are compound 43k1 interaction with protein, (**C**) is compound 43 with protein 6A93.

**Table 1 molecules-27-07285-t001:** Structures of the azinesulfonamides of cyclic amine derivatives and Ki values.

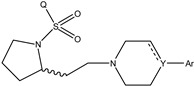					
	**Q**	**Enant**	**Y**	**Ar**	**Ki[nM] ^a^**
					**5-HT**
38	5-Quinolyl	S	N	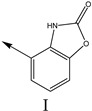	23
39	4-Isoquinolyl				27
40	4-Isoquinolyl	R			36
41	5-Quinolyl	S	N	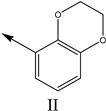	6
42	4-Isoquinolyl				6
43 *	5-Quinolyl	R			5
44	4-Isoquinolyl				7
45	5-Quinolyl	S	C	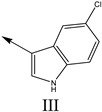	146
46	4-Isoquinolyl				66
47	5-Quinolyl	R			91
48	4-Isoquinolyl				147
49	5-Quinolyl	S	CH	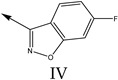	147
50	4-Isoquinolyl				111
51 *	5-Quinolyl	R			120
52	5-Quinolyl	S	N	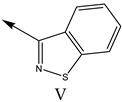	25
53 *	4-Isoquinolyl				37
54	5-Isoquinolyl				37
55	5-Quinolyl	R			24
56	4-Isoquinolyl				14
57	5-Isoquinolyl				16
58 *	5-Quinolyl	S	N	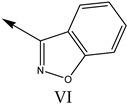	179
59	4-Isoquinolyl				96
60 *	5-Isoquinolyl	R			86
61	5-Quinolyl	S	N	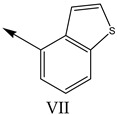	13
62	4-Isoquinolyl				18
63	5-Isoquinolyl				10
64	5-Quinolyl	R			15
65	4-Isoquinolyl				24
66	5-Isoquinolyl				13
67 *	5-Quinolyl	S	N	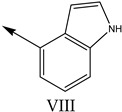	14
68 *	4-Isoquinolyl				8
69 *	5-Quinolyl	R			5
70	4-Isoquinolyl				6
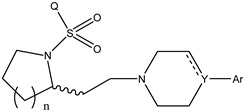					
**Compd.**	**Q**	**N**	**Enant**	**Ar**	**Ki[nM] ^a^**
71	5-Quinolyl	0	S	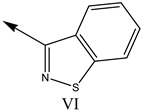	54
72 *	4-Isoquinolyl				56
73	5-Quinolyl	2	R		28
74 *	4-Isoquinolyl				24
75	5-Quinolyl	3	S		48
76	4-Isoquinolyl				50
77	5-Quinolyl	0	S	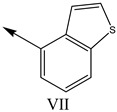	9
78 *	4-Isoquinolyl				16
79	5-Quinolyl	2	R		15
80 *	4-Isoquinolyl				12
81	5-Quinolyl	3	S		12
82	4-Isoquinolyl				8

^a^ Ki values are the means of three independent binding experiments (SEM ≤ 21%), * test set.

**Table 2 molecules-27-07285-t002:** The parameters of the CoMSIA model and the statistical parameters of the best model (shown in bold).

Model	Force Field	q^2^	ONC	r^2^	F Value	SEE
1	S,H and E	0.721	9	0.973	95.423	0.085
2	S,A and E	0.705	9	0.973	95.361	0.085
3	S,D and E	0.633	7	0.972	129.234	0.083
4	S,D and H	0.681	8	0.971	103.102	0.087
5	S,A and H	0.704	8	0.973	111.477	0.084
6	S,A and D	0.558	8	0.972	107.859	0.085
7	E,H and D	0.686	8	0.973	111.177	0.084
8	E,H and A	0.713	8	0.973	111.861	0.083
9	D,A and E	0.599	6	0.969	141.425	0.086
10	S,E,H and D	0.681	8	0.973	111.241	0.084
11	S,E,H and A	0.714	8	0.973	111.845	0.083
12	S,E,D and A	0.608	10	0.973	82.204	0.087
13	E,H,D and A	0.66	10	0.973	82.353	0.087
14	S,H,D and A	0.656	9	0.973	95.170	0.085
**15**	**S,E,D,H and A**	**0.664**	**10**	**0.973**	**82.344**	**0.087**

**Table 3 molecules-27-07285-t003:** The designed compounds and predicted pKi values.

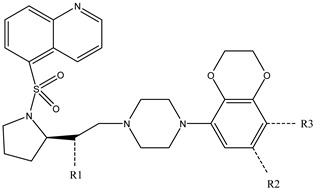					
Compound	R1	R2	R3	pKi	Total score
43a1	CH_2_CH_3_	NH_2_	NO_2_	9.669	9.1321
43b2	CH_2_CH_3_	OH	NO_2_	9.735	8.0002
43c3	CH_2_CH_3_	F	NO_2_	9.856	7.7014
43d4	CHCH_2_CH_3_	NH_2_	NO_2_	8.564	7.3125
43e5	CHCH_2_CH_3_	OH	NO_2_	8.652	7.8187
43f6	CHCH_2_CH_3_	F	NO_2_	8.7	7.5814
43g7	C_6_H_5_	NH_2_	NO_2_	9.63	7.5474
43h8	C_6_H_5_	OH	NO_2_	8.863	7.6443
43i9	C_6_H_5_	F	NO_2_	9.755	7.0177
43j0	NO_2_	NH_2_	C_6_H_5_	9.29	8.4763
43k1	NO_2_	OH	C_6_H_5_	9.554	11.3373
43l2	NO_2_	F	C_6_H_5_	8.843	9.5647
43				8.301	8.2026

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
