# Peer review of "Design and Screening of New Lead Compounds for Autism Based on QSAR Model and Molecular Docking Studies"

_molecules, 2022, doi:10.3390/molecules27217285_

Round 1

Reviewer 1 Report (Previous Reviewer 3)

This research work is interesting in its current form, while i believe it has to be improved:

1-Language errors all over the manuscript please check.

2-Insert supplementary data files as a store for the starting library of your compounds.

3-Add rationale design part in the introduction to justify the selection of these compounds. 

4- conclusion has to be more representative for the whole work.

Author Response

1-Language errors all over the manuscript please check.

We have reviewed and revised the manuscript carefully.

2-Insert supplementary data files as a store for the starting library of your compounds.

Yes, we will submit our compounds by supplementary data file.

3-Add rationale design part in the introduction to justify the selection of these compounds. 

We have provide the rationale design part in the introduction.

4- conclusion has to be more representative for the whole work.

We have rewritten conclusion.

Reviewer 2 Report (New Reviewer)

In this manuscript, the authors performed a simple 3D-QSAR (specifically the CoMSIA) study of a series of azinesulfonamides of cyclic amine derivatives as potential antipsychotics and designed some structures on the basis of the CoMSIA model. However, after careful reading, I do not think this could be published on Molecules.

Some issues worthy of reconsideration:

1.      In the 3D-QSAR studies, the CoMFA model is usually established together with the CoMSIA model to analyze the structure-activity relationship of these compounds in a complementary way. Why the CoMFA analysis is excluded from your study? I suppose the reason that CoMFA only has stereoscopic and electrostatic force fields is insufficient and the analysis merely from the CoMSIA model is not enough to conclude anything. At least you should provide the statistical results of the constructed CoMFA model then I can judge the reliability of your model.

2.      Figure 1 shows the linear correlation diagram of the experimental pKi against the predicted pKi of the test and training set molecules but the fitting is not very good for the test set molecules with highly dispersive points on the plot. How does this indicate that the model has a good predictive ability? The authors are supposed to explain this to readers.

3.      The names of the charts and tables are too casual and confusing to be published. For example, in Figure 1, what do you mean by "A drawing of the predicted pKi and the pKi values (nM) CoMSIA model of the experiment"? In Figure 2, the expression "CoMSIA profile is the compound with the highest affinity 43" is unclear. It is suggested that the names of all charts and tables should be clearly defined and revised. These names are currently unacceptable.

4.      The quality of the structures in Table 1 needs to be improved. To sketch these structures using the same method or software may help.

5.      The quality of Figure 2 is suggested to be improved to see the groups corresponding to different contours more clearly.

6.      Line 180, you mentioned that more than 100 structures were designed on the basis of the QSAR model. I wonder how you designed these structures and what the structure-activity relationship is behind this. What is the concept or principle of your design? The information merely from the CoMSIA model is not enough.

7.      A very important issue is that, from the point of methodology, the CoMSIA model alone can not be used to predict the activity of the designed compounds and to illustrate that some compounds may have good actual activity. The structural features concluded from the CoMSIA model can demonstrate which group favors the activity and serve as guidance for structural design, however, effective prediction and reference of compound activity require detailed study of the compound structures using the Topomer CoMFA method which can realize the fragment-based structure-activity relationship analysis and activity prediction.

8.      The molecular docking analysis is too brief. The docking result of only one compound is not enough to explain the binding mode of this class of compounds. A comparison of compound 43 and the best-scored compound may help.

9.      The conclusion is also too brief and fails to show the significance of this study.

10.  The authors are strongly advised to do some language-editing work since there are lots of grammatical errors and nonstandard expressions in the manuscript. For example, the expressions in the sentences "Basis on the model contour maps…" and "The dock results…" need to be corrected in the abstract. I would say "Based on the model…" and "The docking results…". Line by line careful reading and polishment are necessary.

Author Response

  1. In the 3D-QSAR studies, the CoMFA model is usually established together with the CoMSIA model to analyze the structure-activity relationship of these compounds in a complementary way. Why the CoMFA analysis is excluded from your study? I suppose the reason that CoMFA only has stereoscopic and electrostatic force fields is insufficient and the analysis merely from the CoMSIA model is not enough to conclude anything. At least you should provide the statistical results of the constructed CoMFA model then I can judge the reliability of your model.

 I have added CoMFA analysis, the results were provided in section 2.1.

  1. Figure 1 shows the linear correlation diagram of the experimental pKi against the predicted pKi of the test and training set molecules but the fitting is not very good for the test set molecules with highly dispersive points on the plot. How does this indicate that the model has a good predictive ability? The authors are supposed to explain this to readers.

From table 1, we can find that the 60th compound has the structures of 5-Isoquinolyl benzo[d]isoxazole and enant R, the 72th and 53th compounds have the structures of 4-Isoquinolyl benzo[d]isothiazole and enant S. To our knowledge, chemical structure determines chemical properties. QSAR model shows the relationship of structure with properties. Therefore, for the different structural features of 60th, 72th and 53th compounds with the others, the predict pKi values have large variation (Figure 1). To judge one model, q2, r2, SEE, F values are important. Our model result shows  q2, r2, SEE, F values were 0.664, 0.973, 0.087 and 82.344, respectively, especially value is 0.537. Therefore, our model has a good predictive ability.  

  1. The names of the charts and tables are too casual and confusing to be published. For example, in Figure 1, what do you mean by "A drawing of the predicted pKi and the pKi values (nM) CoMSIA model of the experiment"? In Figure 2, the expression "CoMSIA profile is the compound with the highest affinity 43" is unclear. It is suggested that the names of all charts and tables should be clearly defined and revised. These names are currently unacceptable.

We have revised according to your comments. “Figure 1. The predicted and the experimental pKi values (nM) of CoMSIA model. Table 1. Structures of the azinesulfonamides of cyclic amine derivatives and Ki values. Figure 2. Contour maps of CoMSIA model baised on 43th structure. Table 3 The designed compounds and predicted pKi values.” 

  1. The quality of the structures in Table 1 needs to be improved. To sketch these structures using the same method or software may help.

All the structures in Table 1 have been redrawn in the chemdraw software 8.0. 

  1. The quality of Figure 2 is suggested to be improved to see the groups corresponding to different contours more clearly.

I have improved quality of Figure 2, now it is good.

  1. Line 180, you mentioned that more than 100 structures were designed on the basis of the QSAR model. I wonder how you designed these structures and what the structure-activity relationship is behind this. What is the concept or principle of your design? The information merely from the CoMSIA model is not enough.

CoMSIA can provide stereoscopic, electrostatic, hydrophobic, hydrogen bond donors and hydrogen bond recipients force fields. The 43th compound has the best Ki value, so it was chosen as the template molecule. We can modify 43th structure at the five force fields position. Lots of new structures can be found. Then all the new compounds were predicted pKi values by our QSAR model. The lowest Ki value structure is the best one which was selected to do binding with protein. According to the QSAR model, new structure 43k1 contained the hydrophobic group benzene. And the acceptor group F was used to modify the structure, two H-bonds were found between F and residues ASP-155 and SER-159. Therefor, 43k1 compound has good binding with protein (PDB code: 6A93). 

All the designed new structures will submit to journal together.

  1. A very important issue is that, from the point of methodology, the CoMSIA model alone can not be used to predict the activity of the designed compounds and to illustrate that some compounds may have good actual activity. The structural features concluded from the CoMSIA model can demonstrate which group favors the activity and serve as guidance for structural design, however, effective prediction and reference of compound activity require detailed study of the compound structures using the Topomer CoMFA method which can realize the fragment-based structure-activity relationship analysis and activity prediction.

 I have added the Topomer CoMFA analysis in 2.4 section.

  1. The molecular docking analysis is too brief. The docking result of only one compound is not enough to explain the binding mode of this class of compounds. A comparison of compound 43 and the best-scored compound may help.

We have added the dock result of compound 43 in 2.5 section (Table 3 and Figure 3).

  1. The conclusion is also too brief and fails to show the significance of this study.

 We have revised conclusion.

  1. The authors are strongly advised to do some language-editing work since there are lots of grammatical errors and nonstandard expressions in the manuscript. For example, the expressions in the sentences "Basis on the model contour maps…" and "The dock results…" need to be corrected in the abstract. I would say "Based on the model…" and "The docking results…". Line by line careful reading and polishment are necessary.

         We have careful reading and polished manuscript carefully.

Reviewer 3 Report (New Reviewer)

Minor concerns

There are a lot of grammar errors which should be addressed. A few of the grammar errors are listed below. There are more grammar errors not listed below. It is recommended that the authors revise the manuscript and have it proofread before it was submitted again.

Page 1, second generation significant affinity antipsychotics should be second generation antipsychotics with high affinity

Page 1, This kinds of 5-HT2ARs prominent feature are an additional antagonism (or, more precisely, a high 5-HT2A/D2 receptor affinity ratio), which contributes to the side effects reduction, please revise this sentence. It doesnt make sense.

Page 2, To found should be To find

Page 2, method, it has be used should be method has be used. In addition, calling QSAR as an artificial intelligence is questionable.

Page 2, However, there should be There. No need for However.

Page 2, plays role of should be plays a role in

Page 2, a weak influence on frequency of necrotic effect, please revise. It doesnt make sense to read influence on frequency …”

Page 2, can aided us should be can aid us

Page 2, new structure 43k1 added the hydrophobic group benzene. Should be new structure 43k1 contained the hydrophobic group benzene. Is 43kl a new structure or a published structure? What do you mean a new structure?

Page 2, a satisfied structure was obtained did not make sense. Please revise. How would you define a satisfied structure?

Page 3, we can found should be we can find

Page 3, the structure determines nature. What does it mean?

Page 10, Koll man should be Kollman

Page 8, Table 3, R1, CH-CH3 should be CH2CH3, C6H6 should be C5H6; and R3, C6H6, should be C6H5.  Is the pKI in Table 3 predicted value? Whats the difference between pKI and Total Score? What does it mean for the Total Score?

Page 8, the dock result and affinity prediction are very satisfied. How do you define satisfied?

Major concerns:

For the docking, how will the authors validate the docking method? How about comparing the docking scores to the experimentally available Ki of reported antipsychotics such as the bound structure risperidone and so on?

For the CoMSIA modeling, please show the alignment picture of aligned structures. And provide the training set and test set data in separate tables. Are there any agreement between the docking model and the CoMSIA model?

Author Response

There are a lot of grammar errors which should be addressed. A few of the grammar errors are listed below. There are more grammar errors not listed below. It is recommended that the authors revise the manuscript and have it proofread before it was submitted again.

Page 1, “second generation significant affinity antipsychotics” should be “second generation antipsychotics with high affinity”

Revised.

Page 1, “This kinds of 5-HT2ARs prominent feature are an additional antagonism (or, more precisely, a high 5-HT2A/D2 receptor affinity ratio), which contributes to the side effects reduction”, please revise this sentence. It doesn’t make sense.

We have changed according to your comments.

Page 2, “To found” should be “To find”

Changed.

Page 2, “method, it has be used” should be “method has be used”. In addition, calling QSAR as an artificial intelligence is questionable.

It has been revised.

Page 2, “However, there” should be “There”. No need for However.

Deleted.

Page 2, “plays role of” should be “plays a role in”

Revised.

Page 2, “a weak influence on frequency of necrotic effect”, please revise. It doesn’t make sense to read “influence on frequency …”

We have revised, thank you.

Page 2, “can aided us” should be “can aid us”

Revised.

Page 2, “new structure 43k1 added the hydrophobic group benzene.” Should be ” new structure 43k1 contained the hydrophobic group benzene.” Is 43kl a new structure or a published structure? What do you mean “a new structure”?

I have revised according to your comments. Here, 43k1 is a new structure which was modified basis on 34k structure.

Page 2, “a satisfied structure was obtained” did not make sense. Please revise. How would you define a “satisfied” structure?

I have revised as “Therefor, 43k1 compound has good binding with protein (PDB code: 6A93).”.

Page 3, “we can found” should be “we can find”

We have revised.

Page 3, “the structure determines nature.” What does it mean?

We have revised as “To our knowledge, chemical structure determines chemical properties. QSAR model shows the relationship of structure with properties. ”.

Page 10, “Koll man” should be “Kollman”

Changed.

Page 8, Table 3, R1, CH-CH3 should be “CH2CH3”, C6H6 should be C5H6; and R3, C6H6, should be C6H5.  Is the pKI in Table 3 predicted value? What’s the difference between pKI and Total Score? What does it mean for the “Total Score”?

We have revised according to your comment. SYBYL software has its own scoring function Total Score, which can be used to evaluate the results of each molecular docking. The pKi reflects the tight combination between inhibitor and protein.

Page 8, “the dock result and affinity prediction are very satisfied.” How do you define “satisfied”?

The dock result has higher score and has good interaction of compound with protein, at the same time, predicted Ki is lower than the bast one in our study, which is the satisfied results.

Major concerns:

For the docking, how will the authors validate the docking method? How about comparing the docking scores to the experimentally available Ki of reported antipsychotics such as the bound structure risperidone and so on?

We have added the dock result of compound 43 in 2.5 section (Table 3 and Figure 3) to compare our designed 43k1 structure.

For the CoMSIA modeling, please show the alignment picture of aligned structures. And provide the training set and test set data in separate tables. Are there any agreement between the docking model and the CoMSIA model?

We have added the alignment picture of aligned structures. The test set data has been marked with an asterisk in Table 1. Based on the CoMSIA model, we designed new structures. After predicting Ki values, the best one structures of 43k1 was chosen to do docking study.

Round 2

Reviewer 2 Report (New Reviewer)

NO

Author Response

thank you

Reviewer 3 Report (New Reviewer)

In this revision, the authors have addressed the validation issues, however, there are still a lot of typos and grammers. Most of these errors can be easily taken care of by "Review | Spelling & Grammar", from the first two pages, the following errors have been found, I did not read further since there are so many errors. Please use spelling check to check the errors and resubmit the manuscript.

Page 1, Introduction, “Now days” should be “Nowadays”

Page 1, “bring side effects” should be “bring forth side effects”

Page 1, “affinit” should be “affinity”

Page 2, delete “as an artificial intelligence” after “The Quantitative Structure-Affinity Relationship (QSAR)”, same sentence, “has be used” should be “has been used”

Please note, after idenitying so many errors in the first two pages, the reviewer has lost interest in reading the manuscript further. It is the authors' responsibility to make sure the manuscript is free of typos and grammar errors. 

 Separate issue: 

Page 9, Table 3,

In molecular formula, the number should be in subscript, CH2CH3 should be CH2CH3. Please make changes of functional groups in Table 3 to proper format.

Author Response

  1. Page 1, Introduction, “Now days” should be “Nowadays”

Revised.

  1. Page 1, “bring side effects” should be “bring forth side effects”

We revised it, thank you.

  1. Page 1, “affinit” should be “affinity”

Changed.

  1. Page 2, delete “as an artificial intelligence” after “The Quantitative Structure-Affinity Relationship (QSAR)”, same sentence, “has be used” should be “has been used”

According to your comment, we revised it.

  1. Please note, after idenitying so many errors in the first two pages, the reviewer has lost interest in reading the manuscript further. It is the authors' responsibility to make sure the manuscript is free of typos and grammar errors. 

We have reviewed all paper carefully, and revised the all grammar errors.

  1. Page 9, Table 3,In molecular formula, the number should be in subscript, CH2CH3 should be CH2CH3. Please make changes of functional groups in Table 3 to proper format.

According to your comment, we revised all molecular formulas in Table 3.

This manuscript is a resubmission of an earlier submission. The following is a list of the peer review reports and author responses from that submission.

Round 1

Reviewer 1 Report

This manuscript by Chen and co-workers presents a satisfactory model for predicting pro-social and pro-cognitive effects on azinesulfonamides of cyclic amine derivative as potential anti-psychotics. In my opinion, this topic will be of interest to a wide range of readers who work with the autism and drug design. However, my enthusiasm for this manuscript is tempered by the fact that it needs clarify some information and a complete correction of grammar and language. I suggest that the authors use the Language Editing Services or similar to evaluate accordingly. I picked out some but there are too many corrections to list individually. Some suggestions can be taken into account for the acceptance of the work.

My main criticism is that the work is not well solidified with many validations needed. In the docking results was shown only one pose from the a 43k1 compound. The parameters for choosing this compound were not clear in the text. The result indicates that the 43k1 compound is a good compound because it binding with polymer proteins. Which proteins? What is the benefit of this binding?

Other comments are as follows:

Line 26 - The first paragraph of the introduction has no reference.

Line 36 - The thereby for this disease is mainly depends on drugs… I didn't understand the meaning of the sentence.

Line 36 – Nowdays … spelling mistake. 

Line 37 - dopamine D2 receptors … Are Dopamine D2 receptors target to autism drugs? This is not made clear in the text.

Line 51 - quantitative structure affinity relationship… Quantitative Structure-Affinity Relationship

Line 57 - The quantitative structure affinity relationship (QSAR) as an artificial intelligence method, there are a few studies that were performed in order to find antipsychotics… I didn't understand the meaning of the sentence.

On table 1 the Ki[nM]a is nanomolar but in methods is M. Which is it correct? In Ki[nM]a where do I find the meaning for a? Why did the compounds start with 38 instead of 1? In Q – The name must be changed to the structure group.

In the Figure 1, What is the unity of pKI?

Line 109 - A drawing of the predicted pKI value and the pKI value CoMSIA model of the s experiment. … s?

Line 138 - And free combination, compound modification and optimization selected 12 molecules higher than compound 43 (43a-43k)… 12 or 15? The table 2 contain 15 compounds. 

Line 191 - The Ki values 191 in the micromolar (lM) range were converted to the molar (M) range … micromolar (µM) range?? 

Line 194 – I think that training compounds could be identified on the table 1 with asterisk for example.

Line 209 - 43th instead of 43th. 

Line 209 - We used this method to align all compounds with the 43th because it has a high affinity with 5-HT1A in all compounds... I didn't understand the meaning of the sentence.

Line 240 - Molecular docking was carried out in software Sybyl. We searched RSCB protein database (PDB ID: 6A93) and found that the structure of the ligand in 6A93 is similar to that of our modified compound the 43th. 

There are at least four ligands in the structure 6A93. What is the ligand? 

What is the importance of this information here?

Line 241- We searched RCSB protein database… instead of RSCD.

Line 249- Then, an affinity pocket is created… How?

Line 253 - In our study, molecular docking has been employed to identify a potential binding mode for these inhibitors at 6A93 affinity site and the best docked conformation of molecule 43 was used as template for alignment. Why are you cited the PDB ID instead of protein name? 

Reviewer 2 Report

To be accepted, the authors have to do:

1) clarify the text among lines 93-96;

2) explain the outliers of figure 1;

3) energy of all docked compounds;

4) improve the discussion of figure 3 and compare with the results of figure 2;

5) line 192, is molar or nanomolar?;

Reviewer 3 Report

1-Language errors are many, e.g. line 13 (was), line 14 (on), line 18 and 19 (the highest activity structure) please correct to (the most active compounds) and so on all over the manuscript.

2-The title is not well written please correct it to (Design and screening of new lead compounds for autism Based on QSAR model and molecular docking studies )

3-line 88 (were applied to analyze.) to analyze what please complete the sentence.

4-On what basis you have selected this chemical scaffold specifically, please add the rationale design in the introduction to support your choice.

5-you have selected 45 common skeleton compounds to develop the model, on what basis?

6-please merge results and discussions in one section.

7-conclusion is not well written at all please rewrite  a representative one that reflect all final points you get from your study.

8-figure captions are not appropriate at all e.g. figure 3, (Figure 3. The result of molecular docking.) please try to use more scientific description 

9-the 43k1 compounds, is it one compound or compounds?

10- you have stated that 12 compounds are promising from QSAR studies so why you have performed docking for only one compound?

11-why you did not mentioned validation process for docking studies? 

12-please compare docking studies with references.